# Dissecting genetic and sex-specific sources of host heterogeneity in pathogen shedding and spread

Jonathon A. Siva-Jothy[1], Pedro F. Vale[1,2]*

1 Institute of Evolutionary Biology, School of Biological Sciences, University of Edinburgh, Edinburgh, United Kingdom, 2 Centre for Immunity, Infection and Evolution, University of Edinburgh, Edinburgh, United Kingdom

* pedro.vale@ed.ac.uk

**Data Availability Statement:** All relevant data are within the manuscript and its Supporting Information files.

**Funding:** J.A.S-J was funded by a NERC E3 DTP PhD studentship awarded to the University of

## Abstract

Host heterogeneity in disease transmission is widespread but precisely how different host traits drive this heterogeneity remains poorly understood. Part of the difficulty in linking individual variation to population-scale outcomes is that individual hosts can differ on multiple behavioral, physiological and immunological axes, which will together impact their transmission potential. Moreover, we lack well-characterized, empirical systems that enable the quantification of individual variation in key host traits, while also characterizing genetic or sex-based sources of such variation. Here we used *Drosophila melanogaster* and Drosophila C Virus as a host-pathogen model system to dissect the genetic and sex-specific sources of variation in multiple host traits that are central to pathogen transmission. Our findings show complex interactions between genetic background, sex, and female mating status accounting for a substantial proportion of variance in lifespan following infection, viral load, virus shedding, and viral load at death. Two notable findings include the interaction between genetic background and sex accounting for nearly 20% of the variance in viral load, and genetic background alone accounting for ~10% of the variance in viral shedding and in lifespan following infection. To understand how variation in these traits could generate heterogeneity in individual pathogen transmission potential, we combined measures of lifespan following infection, virus shedding, and previously published data on fly social aggregation. We found that the interaction between genetic background and sex explained ~12% of the variance in individual transmission potential. Our results highlight the importance of characterising the sources of variation in multiple host traits to understand the drivers of heterogeneity in disease transmission.

## Author summary

Host heterogeneity in pathogen transmission presents a major hurdle to predicting and minimizing the spread of infectious agents. Part of the difficulty in linking individual variation to epidemic outcomes is that individual hosts can vary on multiple behavioral, physiological, immunological axes that may affect their transmission potential. Moreover, we

Edinburgh. P.F.V was supported by a Branco Weiss fellowship (https://brancoweissfellowship.org/) and a Chancellor's Fellowship (School of Biological Sciences, University of Edinburgh). The funders had no role in study design, data collection and analysis, decision to publish, or preparation of the manuscript.

**Competing interests:** The authors have declared that no competing interests exist.

lack well-characterized empirical systems that allow to measure multiple facets of individual variation in pathogen transmission. In this work, we capitalize on the strengths of the fruit fly Drosophila as an established and powerful model system for genetics, behavior, and immunity. We provide individual-level data on several axes of infection and test how each of these components experiences variation arising from host genetic background, sex, and mating status. We are therefore able to identify the sources of host heterogeneity (i.e., genetic background, sex) and the specific host traits (social aggregation, pathogen shedding, infection duration) that are most important in determining disease dynamics. We find that a substantial proportion of between-individual heterogeneity in disease transmission is explained by genotype-by-sex interactions affecting the likelihood that individuals will shed virus, but also how much they are likely to shed.

## Introduction

Individual host heterogeneity in pathogen transmission is a pervasive feature of all host-pathogen systems [1–5]. Such heterogeneity is so common that it has been generalised into the '20–80 rule' because of the frequent observation that 20% of hosts contribute to roughly 80% of transmission [1, 5, 6]. More extreme forms of heterogeneity can result in rarer 'superspreading' events capable of causing large outbreaks of infectious disease in human and animal populations [2, 7]. Mary Mallon, who became known as 'Typhoid Mary', was a superspreader of particular infamy, infecting over 50 people with *Salmonella typhi* while working as a cook in New York during the early 20th century [8]. Beyond this anecdotal example, the 2003 outbreaks of SARS in Singapore and Hong Kong were greatly accelerated by a few superspreading individuals who caused over 70% of all SARS transmission [9]. Similar levels of individual heterogeneity in transmission were recorded and are thought to have accelerated the spread of MERS, Ebola and most recently, SARS-CoV2 [10, 11].

Outbreaks of infectious disease are often difficult to predict, especially when the effect of individual heterogeneity is difficult to assess using traditional measures of outbreak risk. A widely used metric for the rate of pathogen spread is the basic reproductive number, $R_0$, which estimates the average number of expected secondary infections caused by a single infected individual in a completely susceptible population. While this metric is extremely useful as a broad scale predictor of epidemic spread (or fadeout), by focussing on the population average, $R_0$ conceals outliers with a potentially higher propensity for pathogen spread [2, 3, 12]. A clearer understanding of what drives heterogeneity in pathogen transmission requires a framework capable of accounting for such between-individual variation, which could enable more efficient control strategies that specifically target and treat high-risk individuals [2, 4]. Indeed, several empirical studies in human and non-human pathogens have found that accounting for transmission heterogeneity in disease interventions can have a disproportionally large effect on curtailing pathogen transmission [1, 13, 14]. The importance of predicting high-risk individuals before outbreaks occur has therefore pushed understanding the causes of heterogeneity in disease transmission to the forefront of epidemiology and disease ecology research [3, 12, 15, 16].

Despite being common, the underlying causes of heterogeneity in pathogen transmission remain elusive. Individual variation in host contact networks may be an important factor: it was Typhoid Mary's position as a cook which exposed her to so many susceptible individuals. However, what enabled Typhoid Mary to stay in this role was her status as an asymptomatic carrier of the infection, which led to her release from quarantine on several occasions [8].

Similarly, the absence of symptoms in a number of SARS superspreaders delayed their admission to hospital and allowed them to continue spreading the virus [17]. These examples underline that achieving a detailed understanding of the sources of heterogeneity in pathogen transmission is challenging because it results from complex interactions between multiple host traits affecting social interactions and host-pathogen dynamics. By dissecting the underlying genetic or environmental sources of variation in these traits we can begin to assess how they influence three key components of pathogen transmission: i) contact rate between infected and susceptible individuals; ii) the likelihood that contact will result in infection; and iii) for pathogens with continuous transmission, the duration of infection [12].

Infected-susceptible host contact rate is heavily influenced by host behaviours affecting locomotion and aggregation, and may be affected by population density [18], social group size [19], and behavioural syndromes [20]. Social networks often exhibit extreme heterogeneity in the wild [21, 22] and factors such as host genotype, sex condition, age and personality are known to affect social aggregation in numerous species [23–26]. Once individuals acquire an infection, their ability to clear and shed pathogens is mainly determined by physiological and immune mechanisms. Variation in these mechanisms primarily influences the likelihood of pathogen transmission and the duration of infection [12, 27]. Many genetic and environmental sources of variation in physiological immunity have been described [28–30] including coinfection [31, 32], nutrition [33, 34], and stress [35, 36]. It is relevant to note however that most work has addressed the isolated effects of behavioural, physiological and immune traits on transmission. While phenotypic variation in these traits is not surprising, the way in which these traits co-vary and how this covariation may drive pathogen transmission is less understood [4, 37–39]. Extreme host heterogeneity in transmission is therefore likely to be the outcome of behavioral traits that promote increased contact co-occurring with physiological and immune traits that increase infectiousness [37–40]. Crucially, it is important not only to quantify and measure the effect of heterogeneity in single traits that may facilitate transmission, but it is key to understand how multiple traits co-vary as 'coupled heterogeneities'[4, 39–43]. To further understand the sources of heterogeneity in pathogen transmission and how they may co-vary, it is therefore important to measure the causes and the consequences of variation in multiple behavioural, physiological, and immune traits.

In the present work we aimed to test how common sources of variation between individuals—genetic background, sex and mating status—contribute to individual heterogeneity in pathogen transmission potential. The fruit fly, *Drosophila melanogaster*, is a powerful and genetically tractable model of infection, immunity and behaviour [44–46]. Despite the widespread use of *D. melanogaster* to address questions of immunity and behavior, it is not commonly used to combine these strengths as a model of experimental epidemiology (but see [47, 48]). Here, we seek to leverage the strengths of the *D. melanogaster* system to explore how variation in key behavioral and physiological traits may impact individual disease transmission potential.

We infected males and females from a range of naturally derived *D. melanogaster* genotypes with *Drosophila* C Virus (DCV). DCV is a horizontally transmitted fly pathogen which causes behavioural, physiological and metabolic pathologies [26, 49–52]. Relatively little is known about DCV infection in wild Drosophila, where it appears to be rare (surprisingly so for a model viral infection of fruit flies)[53]; however *Drosophila* lab stocks are often found to have low-level persistent DCV infections[53, 54], suggesting that horizontal transmission–which is assumed to be fecal-oral—is common. Cannibalism of infected fly cadavers has also been shown to be a viable route of transmission[55], but it is not clear if this is a common transmission route in the wild.

We measured four host traits and infection outcomes that directly impact pathogen transmission. In a first experiment, we measured (1) the infected lifespan and (2) the viral load at death (VLAD) in the same flies. The infected lifespan is relevant in the context of transmission because it is the time during which pathogen shedding can occur. The VLAD is a useful measure because some transmission may still be possible following fly death[55]. In a second experiment, we measured (3) the internal viral load, and (4) how much virus was shed, by the same fly, on either the first-, second- or third-day following infection. Measuring the quantity of virus growing inside a fly and the quantity shed provides important insight into how the host's ability to limit pathogen proliferation may affect host infectiousness. Furthermore, measuring viral load and shedding during the first three days of infection helps to characterize early stage host-pathogen dynamics during which time DCV is actively growing[56]. Finally, we integrated these measurements, in addition to previously described data on variation in social aggregation [26] into a composite metric of individual transmission potential, $V$ [2, 12]. Examining $V$ allowed us to assess how genetic and sex-specific variation may generate between-individual heterogeneity in pathogen transmission.

## Methods

### Flies and rearing conditions

Flies used in experiments were 3–5 days old and came from ten lines of the *Drosophila* Genetic Resource Panel (DGRP). These genetic backgrounds are five of the most resistant (RAL-379, RAL-59, RAL-75, RAL-138, RAL-502) and susceptible (RAL-765, RAL-818, RAL-380, RAL-373, RAL-738) to systemic Drosophila C Virus infection [57], and were chosen to provide a wide range of susceptibility to the virus. Virgin females were isolated from males within 7 hours of eclosion. Mated females and males were produced by rearing one female with one male for 24 hours. Mating was confirmed by observing oviposition within the following 24 hours and these egg's subsequent development. Flies were reared in plastic vials on a standard diet of Lewis medium at 18±1°C with a 12 hour light:dark cycle. Stocks were tipped into new vials approximately every 14 days. One month before the experiments, flies were maintained at low density (~10 flies per vial, to guarantee abundant resources) for two generations at 25±1°C with a 12 hour light:dark cycle.

### Virus culture and infection

The *Drosophila* C Virus (DCV) isolate used in this experiment was originally isolated in Charolles, France and grown in Schneider *Drosophila* Line 2 (DL2) as previously described [52], diluted ten-fold ($10^8$ infectious units per ml) in TRIS-HCl solution (pH = 7.3), aliquoted and frozen at -70°C until required. To infect with DCV, flies were pricked in the pleural suture with a 0.15mm diameter pin, bent at 90˚ ~0.5mm from the tip, dipped in DCV.

### Measuring lifespan and viral load at death

Lifespan and viral load at death were measured in the same fly. Following DCV infection, flies were isolated and reared in standard vials. Flies were then monitored every day until all individuals died, when they were removed from vials, fixed in 50μl of TRI-reagent and frozen at -70°C. For most treatment groups, infected lifespan and viral load at death were measured for n = 17–20 individual flies (S1 Table).

### Viral growth and shedding measurement setup

Due to destructive sampling, we measured the viral load and shedding of single flies at a single time point, either 1-, 2- or 3-days post-infection (DPI). Following DCV infection, single flies

were placed into 1.5ml Eppendorf tubes with ~50µl of Lewis medium in the bottom of the tube. To measure viral shedding, flies were transferred to tubes for 24 hours, immediately following 1 or 2 days after systemic infection. Following a further 24 hours, flies were removed and homogenised in 50µl of TRI-reagent, tubes were also washed out with 50µl of TRI-reagent by vortexing. These samples were then frozen at -70˚C, to await DCV quantification by qPCR. For each combination of sex and genetic background, over the three days, we measured viral load and virus shedding in n = 7–15 flies per fly line / sex combination (S2 Table).

### DCV RNA extraction

To measure viral load at death and viral shedding, RNA was extracted from samples by Phenol-Chloroform extraction. Samples were thawed on ice for 30 minutes before being incubated at room temperature for 5 minutes. Samples were then centrifuged at 12,000×g for 10 minutes at 4˚C after which large debris was removed. For phase separation, samples were shaken vigorously for 15 seconds, 10µl of chloroform added, and incubated at room temperature for a further 3 minutes before being centrifuged at 12,000×g for 15 minutes at 4˚C. Following phase separation, the upper aqueous layer was removed from each sample and added to 25µl of isopropanol, tubes were then inverted twice, before being centrifuged at 12,000×g for 10 minutes at 4˚C. Precipitated RNA was then washed by removing the supernatant, and re-dissolving the RNA pellet in 50µl of 75% ethanol before being centrifuged at 7,500×g for 5 minutes at 4˚C. RNA suspension was achieved by removing 40µl of the ethanol supernatant, allowing the rest to dry by evaporation and dissolving the remaining RNA pellet in 20µl of RNase-free water. To measure viral load after 1, 2 or 3 days of infection, we extracted RNA from flies using a semi-automatic MagMAX Express Particle Processor using the MagMAX-96 total RNA isolation kit manufacturer's protocol [58] with the elution step extended to 18 minutes. RNA samples were stored at -70˚C to await reverse transcription.

### Reverse transcription and qPCR Protocol

Extracted RNA was reverse-transcribed with M-MLV reverse transcriptase and random hexamer primers, before being diluted 1:1 with nuclease free water. cDNA samples were stored at -20˚C to await qPCR analysis. DCV titre was quantified by qPCR using Fast SYBR Green Master Mix in an Applied Biosystems StepOnePlus system. Samples were exposed to a PCR cycle of 95˚C for 2 minutes followed by 40 cycles of: 95˚C for 10 seconds followed by 60˚C for 30 seconds. Forward and reverse primers used included 5'-AT rich flaps to improve fluorescence (DCV_Forward: 5' AATAAATCATAAGCCACTGTGATTGATACAACAGAC 3'; DCV Reverse: 5' AATAAATCATAAGAAGCACGATACTTCTTCCAAACC 3'). Across all plates, two technical replicates were carried out per sample. DCV titre was calculated by absolute quantification, using a standard curve created from a 10-fold serial dilution (1-10$^{-12}$) of DCV cDNA. Our detection threshold was calculated for each plate using the point at which two samples on our standard curve gave the same Ct value. The point of redundancy in a standard curve was taken to be equivalent to 0 viral particles. Due to our detection protocol measuring viral copies of RNA, we cannot comment on the viability of any detected virus. We transformed our measurements of viral RNA in order to obtain the number of virus copies per fly.

### Calculating individual variation in transmission potential, *V*

We used measures of virus shedding, lifespan following infection, and social aggregation to simulate individual transmission potential. We integrated these measures using a simple framework that describes transmission potential as a function of contact rate between susceptible and infected individuals, the likelihood that such contact will result in infection, and the duration of

the infectious period [12]. Using previously analysed data on social aggregation [26], we used nearest neighbour distance as a measure of contact rate. Flies that aggregated more closely to conspecifics, are assumed to have a higher contact rate, and are therefore more likely to spread DCV. We also assume that transmission likelihood increases with virus shedding. We therefore take the amount of virus shed by flies as a proximate measure of the likelihood that contact will result in infection. Using these traits, individual transmission potential, $V$, was calculated as:

$$V = \frac{(\text{Virus Shedding Titre}) \times (\text{Lifespan})}{(\text{Aggregation Distance})}$$

Aggregation distance, lifespan following infection and virus shedding were all measured in separate experiments. Therefore, to calculate $V$ as a measure of individual transmission potential, we simulated theoretical individuals by bootstrapping trait values sampled from each of these three datasets. Values of $V$ were then divided by the population mean, $V_{mean}$ to enable clearer interpretations of how individual transmission potentials relate to the population average (which is itself similar to $R_0$). For example, an individual with a $V/V_{mean}$ of 3, has a transmission potential three times greater than the population average transmission. We simulated 60 individuals for each combination of sex and genetic background, assuming no specific covariance structure between traits, that is, all possible trait combinations were considered.

## Statistical analysis

Across all experiments, generalised linear models (GLMs) were used to analyse continuous response variables and logistic regressions were used to analyse proportions. An effect of sex or mating was analysed in separate models comparing males or virgin females to the same dataset of mated females, respectively.

To analyse lifespan, two GLMs were constructed containing a three-way interaction genetic background, VLAD, and sex or mating (S3 Table). The two GLMs for VLAD, contained either a two-way interaction between genetic background and sex or a two-way interaction between genetic background and mating (S3 Table).

Due to zero-inflation, we used two models to sequentially analyse both viral load and virus shedding data. Viral load and virus shedding are broken down into qualitative (the proportion of non-zero values) and quantitative variation (differences between non-zero values). First, we conducted logistic regressions on all of the values in these datasets and analysed the proportion of values that were greater than zero. Logistic regressions analysing sex-differences in viral load included DPI (a 3-level factor: 1, 2 or 3 days) and an interaction between genetic background and sex (S3 Table). For analysing the effect of mating in females on viral load, logistic regressions included DPI and an interaction between genetic background and mating (S3 Table). Logistic regressions of virus shedding used a similar model that also included quantitative viral load as a predictor (S3 Table). After these logistic regressions, zeroes were removed from all datasets to analyse the subset of positive-values. The GLMs used to analyse these subsets included the same predictors as their corresponding logistic regressions, for viral load: an interaction between genetic background and sex or mating, alongside DPI, with the inclusion of quantitative viral load for virus shedding (S3 Table).

Due to zero-inflation $V$ was also analysed with a logistic regression followed by a GLM. A logistic regression was used to analyse the proportion of $V$ values that were greater than zero with a two-way interaction between sex and genetic background as predictors (S3 Table). Zero-values of $V$ were then removed from the dataset, and a GLM was used to analyse differences in the size of $V$, with an interaction between sex and genetic background included as a predictor (S3 Table).

We calculated the amount of deviance and variance explained by predictors in logistic regressions and GLMs, respectively, by dividing the total deviance or variance explained by the model. Where appropriate, we corrected for multiple testing using Bonferroni correction. All statistical analyses and graphics produced in R 3.3.0 using the *ggplot2* [59], *lme4* [60] and *mult-comp* [61] packages.

## Results

### Lifespan following infection

To begin our characterization of host heterogeneity in pathogen spread we measured fly lifespan following septic infection with DCV. This allowed us not only to confirm that the lines we chose showed a range of continuous variation in susceptibility[57], but infected lifespan is also directly relevant to the individual transmission potential V [12], by determining the full extent of the infectious period (see *Calculating Individual Variation in Transmission Potential*, *V* above). Infected lifespan varied significantly between males and females and the extent of this variation differed between host genetic backgrounds, as expected (Fig 1A and Table 1). Across all genetic backgrounds, males had a mean lifespan of 14.1 days, with RAL-75 (16 days) surviving the longest and RAL-765 (11.1 days) surviving for the shortest amount of time. Females showed a slightly lower average lifespan of 13.06 days, with RAL-59 (15.1 days) and RAL-765 (11.4 days) surviving for the longest and shortest duration of time, respectively. When comparing mated males and females, or virgin and mated females, genetic background explained the most variance of any predictor (7% and 10.9%, respectively; Table 1). We found no evidence that mating affected the lifespan of females following DCV infection (Fig 1A and Table 1).

We also decided to measure the viral load at death (VLAD) in all flies on their day of death. While we expected continuous viral expulsion through defecation to be the main route of pathogen shedding, it is also possible that some transmission happens following fly death, for example via trophic transimssion routes [55]. The VLAD did not differ between genetic backgrounds, sex or female mating status (Fig 1B and Table 2), and flies that died sooner following infection had greater VLAD than those that died later (Fig 1C and Table 1).

### Viral load

Next, in a separate experiment, we quantified the effects of host genetics, sex and mating on host infectiousness, which we considered to include both the ability to control viral growth and the extent to which virus is shed. Following 24 hours of septic DCV infection we observed that a substantial number of flies did not have detectable DCV titres, as measured using qPCR (Figs 2A and S1). Flies with no detectable DCV loads could reflect individuals able to limit initial viral growth or could be caused by viral titres below the detection threshold of our qPCR and therefore reflect individuals with extremely low DCV loads. In a more natural setting, these zero-values would reflect what is observed in recently infected individuals or individuals successfully limiting the growth of the pathogen. Crucially, in the context of controlling disease transmission, individuals with low viral titres are likely to go undetected and may become a transmission risk later if their viral load increases. We first analysed this "qualitative" DCV load reflecting the complete presence or absence of detectable DCV titres (Fig 2A and Table 3). The proportion of flies with detectable DCV titres varied significantly over time (Fig 2A and Table 3), and the predictors that explained the most deviance were genotype-by-sex interactions (when comparing mated males and females) and genotype-by-mating interactions (when comparing mated or virgin females), although each of these effects only accounted for roughly 5% of the deviance (Table 3).

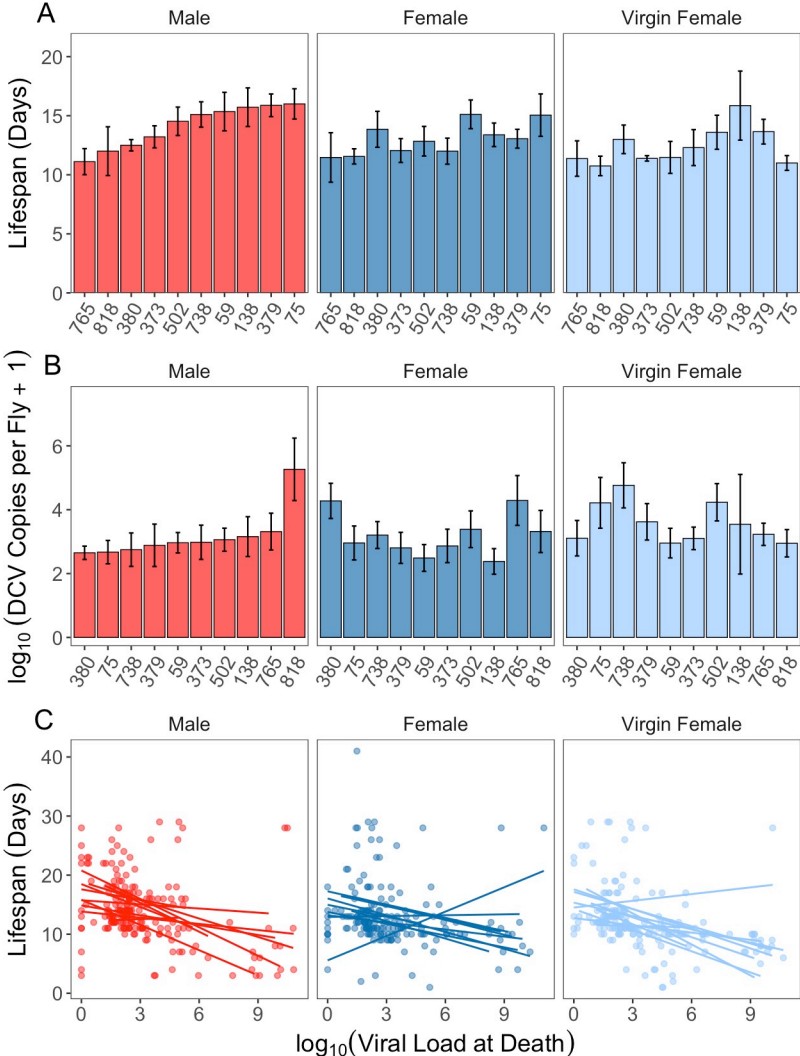

**Fig 1.** Mean±SE (a) lifespan in days following infection and (b) the viral load at death in males (red), mated females (blue), and virgin females (pale blue) of ten genetic backgrounds. The x-axis shows the line number form the DGRP panel and is in ascending order according to male flies. (c) the relationship between lifespan following infection and the viral load of flies at death. Each point is an individual male (red), mated female (blue), or virgin female (pale blue) fly. The nature of this relationship within each genetic background is represented by a line of best fit with outlier backgrounds labelled.

We then focused on quantitative variation in the subset of flies with detectable DCV titres, as a proxy for host variation in the ability to limit viral growth. We saw significant changes in quantitative DCV load variation over the first three days of infection. Viral load peaked 2-days post-infection (pairwise comparison, p = 0.0012), before decreasing to the similar levels as 1-day post-infection at 3-days post-infection (pairwise comparison, p = 0.068). A substantial proportion of the variance in viral loads could be attributed to genotype-by-sex interactions (19.2%, when comparing mated males and females) (Fig 2B and Table 4).

## Virus shedding

During the destructive sampling of flies to quantify internal viral titres, we were also able to obtain paired measures for each fly of the amount of virus shed after one, two, or three days.

**Table 1. Model outputs for the generalized linear modelling tests performed on lifespan following DCV infection.** The VLAD acronym is used in place of 'viral load at death'. Separate analyses were used to test the effect of sex (top model) and mating in females (bottom model).

| Response Variable | Predictor | Df | F | %Variance Explained | p-value |
|---|---|---|---|---|---|
| Lifespan Following Infection | Sex | 1 | 2.00 | 0.6 | 0.16 |
| | Genetic Background | 9 | 3.92 | 10.9 | <0.0001 |
| | VLAD | 1 | 38.9 | 12.1 | <0.0001 |
| | Sex*Genetic Background | 9 | 0.96 | 2.7 | 0.47 |
| | Sex*VLAD | 1 | 5.46 | 1.7 | 0.02 |
| | Genetic Background*VLAD | 9 | 0.63 | 1.8 | 0.77 |
| | Sex*Genetic Background*VLAD | 9 | 2.67 | 7.4 | 0.005 |
| | Mating | 1 | 2.74 | 0.9 | 0.099 |
| | Genetic Background | 8 | 2.43 | 7.0 | 0.01 |
| | VLAD | 1 | 32.3 | 10.2 | <0.0001 |
| | Mating*Genetic Background | 8 | 0.54 | 1.5 | 0.84 |
| | Mating*VLAD | 1 | 3.78 | 1.2 | 0.053 |
| | Genetic Background*VLAD | 8 | 1.71 | 4.9 | 0.087 |
| | Mating*Genetic Background*VLAD | 8 | 1.46 | 4.2 | 0.16 |

Shed virus was measured by housing single flies in an Eppendorf tube for 24 hours, removing flies from their tube, washing the tube out with TRIzol, and quantifying the DCV RNA present using qPCR. Similar to the measurement of viral load, we did not detect DCV in the shedding of a number of flies (Figs 3A and S1). Again, we interpret these zeroes values to be reflective of individuals that shed very low titres of DCV, or no virus at all. Qualitative shedding (the presence or absence of detectable virus shedding) varied significantly over the three days post infection and was affected by genetic background and its interaction with sex (Fig 3A and Table 5). However, while these effects were significant, they explained only 2–3% of the deviance in the proportion of flies with detectable viral shedding. We did not detect a significant effect of mating on the presence or absence or DCV shedding (Fig 3A and Table 5).

In flies where DCV shedding was detected, we compared the amount of virus shed in the vials across the 3 days. Similar to the viral loads measured within the flies, viral shedding also peaked at day 2 in most fly genetic backgrounds (Fig 3B, Tables 5 and 6, pairwise comparisons, p<0.0001). Quantitative variation in DCV shedding was explained by genetic background and the extent of this variation was determined by female mating status, but not sex (Fig 3B and Table 6). The amount of variance explained by sex was <1%, in comparison with genetic background (9.48% and 5.82%) and its interactions with sex (8.87%) or mating (6.53%) (Table 6). Across all treatment groups, we found no significant relationship between viral load and shedding (S1 Fig and Table 6).

**Table 2. Model outputs for the generalized linear modelling tests performed on the viral load at death of flies infected with DCV.** Separate analyses were used to test the effect of sex (top model) and mating in females (bottom model).

| Response Variable | Predictor | Df | F | % Variance Explained | p-value |
|---|---|---|---|---|---|
| Viral Load at Death (VLAD) | Sex | 1 | 0.17 | 0.05 | 0.68 |
| | Genetic Background | 9 | 0.96 | 2.53 | 0.47 |
| | Sex*Genetic Background | 9 | 0.92 | 2.43 | 0.50 |
| | Mating | 1 | 1.90 | 0.57 | 0.17 |
| | Genetic Background | 8 | 1.30 | 3.5 | 0.24 |
| | Mating*Genetic Background | 8 | 0.93 | 2.49 | 0.50 |

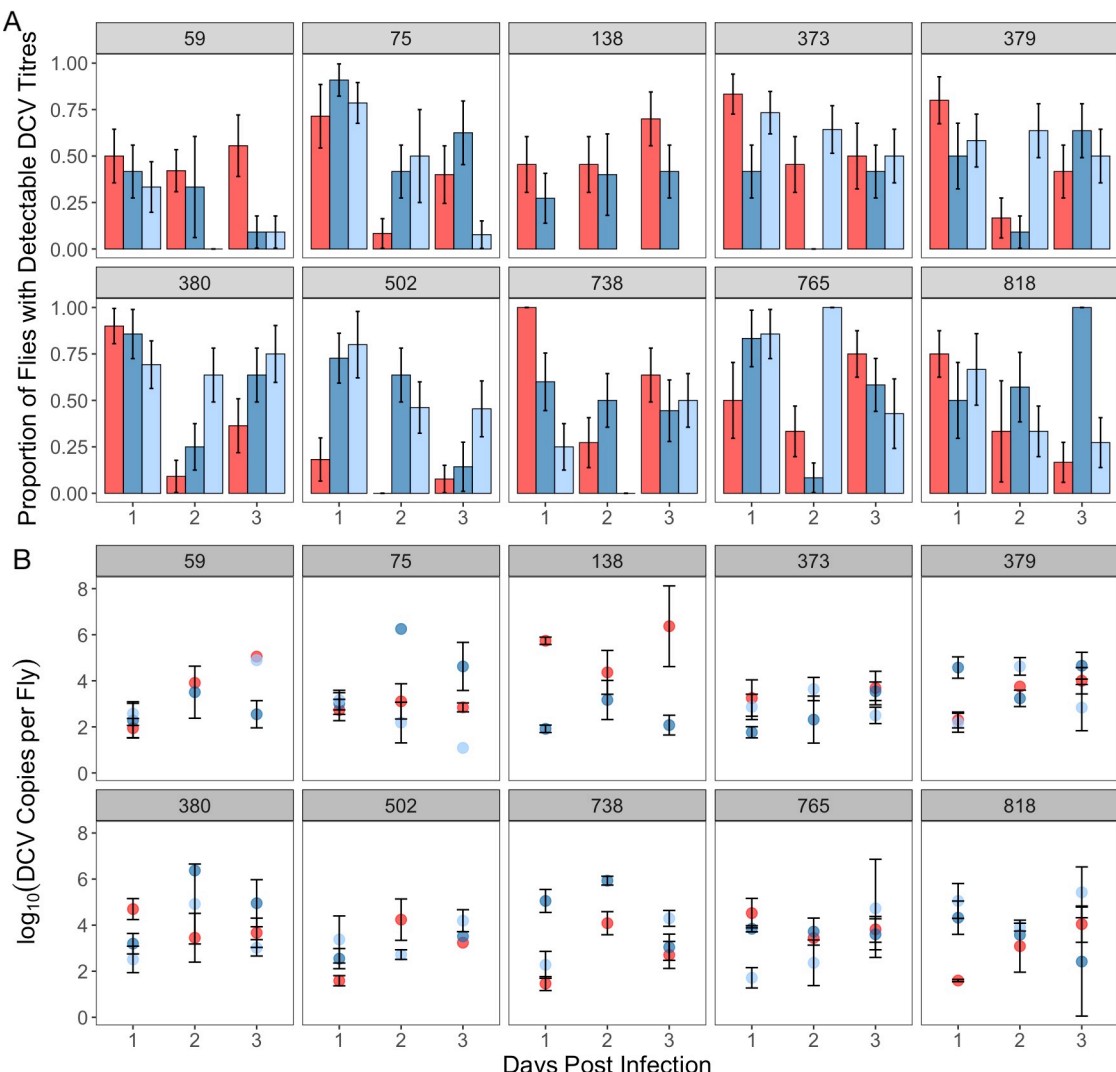

**Fig 2.** Mean±SE (a) proportion of flies with detectable loads of DCV and the (b) viral titre of flies with non-zero DCV loads, over the first 3 days of infection. Across both panels, numbers in each pane denote the genetic background from the DGRP, while the colour of bars, points and lines represent sex and mating status. Males are shown in red, mated females in blue, and virgin females in pale blue.

**Table 3. Model outputs for the binomial logistic regression conducted on qualitative DCV loads (the proportion of non-zero DCV loads).** The DPI acronym is used in place of 'days post-infection'. Separate analyses were used to test the effect of sex (top model) and mating in females (bottom model).

| Response Variable | Predictor | Df | $\chi^2$ | % Deviance Explained | *p*-value |
|---|---|---|---|---|---|
| Qualitative DCV Load (is DCV detectable–yes/no) | Sex | 1 | 0.019 | 0.002 | 0.89 |
| | Genetic Background | 9 | 9.58 | 1.18 | 0.39 |
| | DPI | 2 | 36.6 | 4.52 | <0.0001 |
| | Sex*Genetic Background | 9 | 45.2 | 5.58 | <0.0001 |
| | Mating | 1 | 1.01 | 0.13 | 0.31 |
| | Genetic Background | 8 | 22.4 | 2.83 | 0.008 |
| | DPI | 2 | 27.2 | 3.43 | <0.0001 |
| | Mating*Genetic Background | 8 | 39.0 | 4.92 | <0.0001 |

**Table 4. Model outputs for the GLM analysis conducted on quantitative DCV load (the titres of non-zero DCV loads).** The DPI acronym is used in place of 'days post-infection'. Separate analyses were used to test the effect of sex (top model) and mating in females (bottom model).

| Response Variable | Predictor | DF | F | % Variance Explained | p-value |
|---|---|---|---|---|---|
| Quantitative DCV Load (How much DCV is detected) | Sex | 1 | 0.0062 | 0.003 | 0.94 |
| | Genetic Background | 9 | 2.24 | 7.94 | 0.02 |
| | DPI | 2 | 3.37 | 2.65 | 0.036 |
| | Sex* Genetic Background | 9 | 5.41 | 19.2 | <0.0001 |
| | Mating | 1 | 0.68 | 0.26 | 0.41 |
| | Genetic Background | 8 | 3.18 | 11.0 | 0.0012 |
| | DPI | 2 | 4.66 | 3.60 | 0.01 |
| | Mating* Genetic Background | 8 | 1.42 | 4.38 | 0.19 |

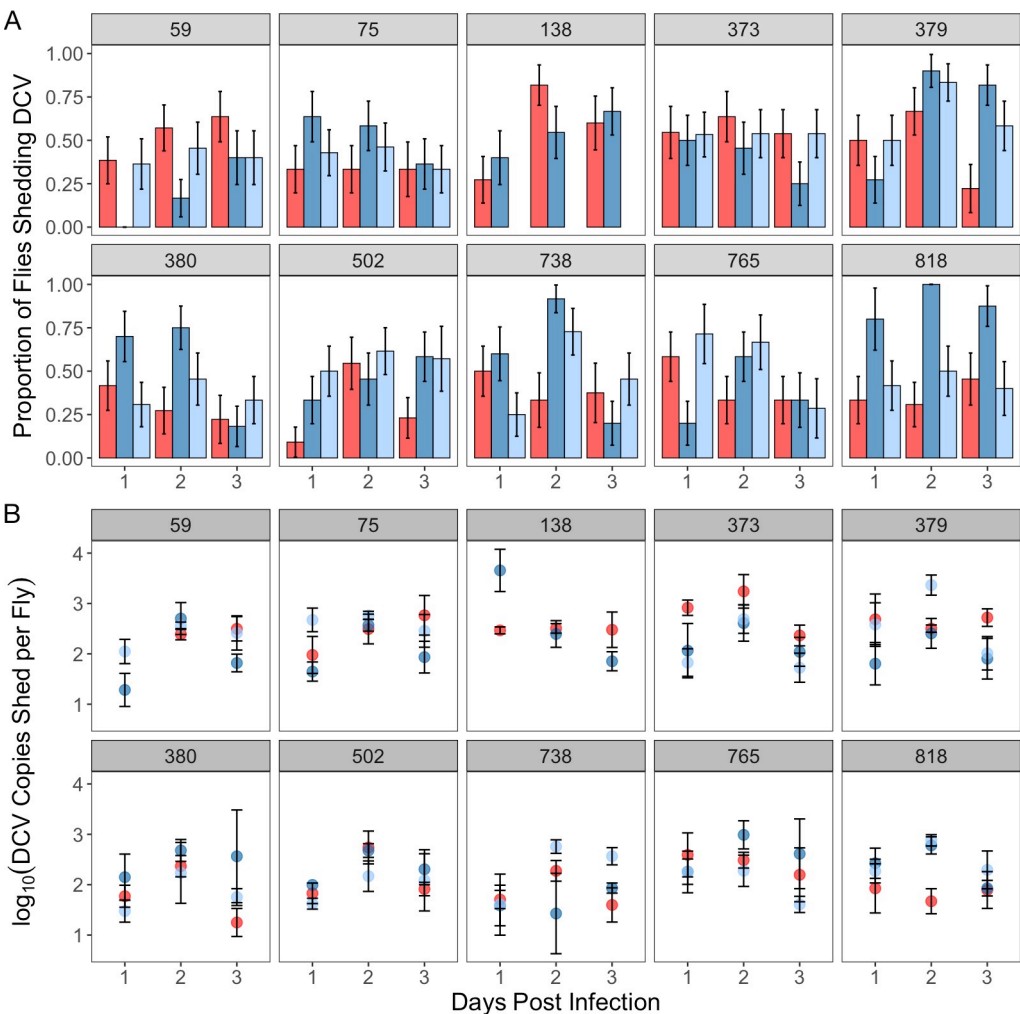

**Fig 3.** Mean±SE (a) proportion of flies shedding non-zero titres of DCV and the (b) titre of the non-zero virus shedding subset over the first 3 days of infection. Panels denote genetic background, while the colour of bars, points and lines represent sex and mating status. Males are shown in red, mated females in blue, and virgin females in pale blue. Bars of the same colour in each in pane in panel (a) represent (from left to right) days 1, 2 and 3 of infection.

**Table 5. Model outputs for the GLM analysis conducted on qualitative DCV shedding (the proportion of sheddings with non-zero readings of DCV).** The DPI acronym is used in place of 'days post-infection'. Separate analyses were used to test the effect of sex (top model) and mating in females (bottom model).

| Response Variable | Predictor | Df | $\chi^2$ | % Deviance Explained | p-value |
|---|---|---|---|---|---|
| Qualitative DCV Shedding (Was any shedding detected–yes/no) | Sex | 1 | 4.93 | 0.64 | 0.026 |
| | Genetic Background | 9 | 17.6 | 2.27 | 0.04 |
| | Viral Load | 1 | 0.03 | 0.004 | 0.85 |
| | DPI | 2 | 25.1 | 3.25 | <0.0001 |
| | Sex*Genetic Background | 9 | 23.8 | 3.07 | 0.005 |
| | Mating | 1 | 1.33 | 0.18 | 0.25 |
| | Genetic Background | 8 | 19.0 | 2.53 | 0.025 |
| | Viral Load | 1 | 1.10 | 0.15 | 0.29 |
| | DPI | 2 | 7.66 | 1.02 | 0.022 |
| | Mating*Genetic Background | 8 | 8.12 | 1.08 | 0.42 |

## Variation in transmission potential, *V*

To generate an estimate of an individual's transmission potential, *V* [2, 12], we combined the single measures of lifespan and virus shedding (1, 2 or 3 DPI) described above with previously published data on social aggregation in males and females of the same genetic backgrounds [26]. By simulating theoretical "individuals" where each trait value was sampled from these experimentally derived distributions of lifespan, shedding and asocial aggregation, we obtained a distribution of the variation in individual transmission potential V (Fig 4A). This distribution of transmission potential, *V*, was zero-inflated (Fig 4A), likely as a result of many flies not shedding DCV (Fi. 3A). Zero values of *V* therefore represent individuals with no transmission risk (Fig 4A), as flies that shed no virus had, by definition, no transmission potential, irrespective of their aggregation and lifespan. The distribution of *V* was also overdispersed, with an extreme right tail, comprised of individuals with high pathogen transmission potential relative to the population average (Fig 4A). Given the zero-inflation of V, we first analysed the 'qualitative' variation in *V* (the proportion of flies where *V*>0, that is, where the likelihood of an outbreak is either zero or positive). By analysing the source of variation in which is reflective of an outbreak likelihood). We found significant differences between males and females in presenting a V>0, with the extent of this difference also determined by genetic background (Fig 4B and Table 7). Sex (0.28%), genetic background (2.3%) and the interaction

**Table 6. Model outputs for the GLM analysis conducted on quantitative DCV shedding (the subset of shedding with non-zero readings of DCV).** The DPI acronym is used in place of 'days post-infection'. Separate analyses were used to test the effect of sex (top model) and mating in females (bottom model).

| Response Variable | Predictor | Df | F | % Variance Explained | p-value |
|---|---|---|---|---|---|
| Quantitative DCV Shedding (How much virus was shed?) | Sex | 1 | 0.67 | 0.28 | 0.42 |
| | Genetic Background | 9 | 2.52 | 9.48 | 0.009 |
| | Viral Load | 1 | 5.03 | 4.21 | 0.007 |
| | DPI | 2 | 0.23 | 0.095 | 0.63 |
| | Sex*Genetic Background | 9 | 1.73 | 6.53 | 0.082 |
| | Mating | 1 | 0.22 | 0.098 | 0.64 |
| | Genetic Background | 8 | 1.44 | 5.82 | 0.17 |
| | Viral Load | 1 | 11.2 | 10.1 | <0.0001 |
| | DPI | 2 | 0.18 | 0.08 | 0.67 |
| | Mating*Genetic Background | 8 | 2.46 | 8.87 | 0.014 |

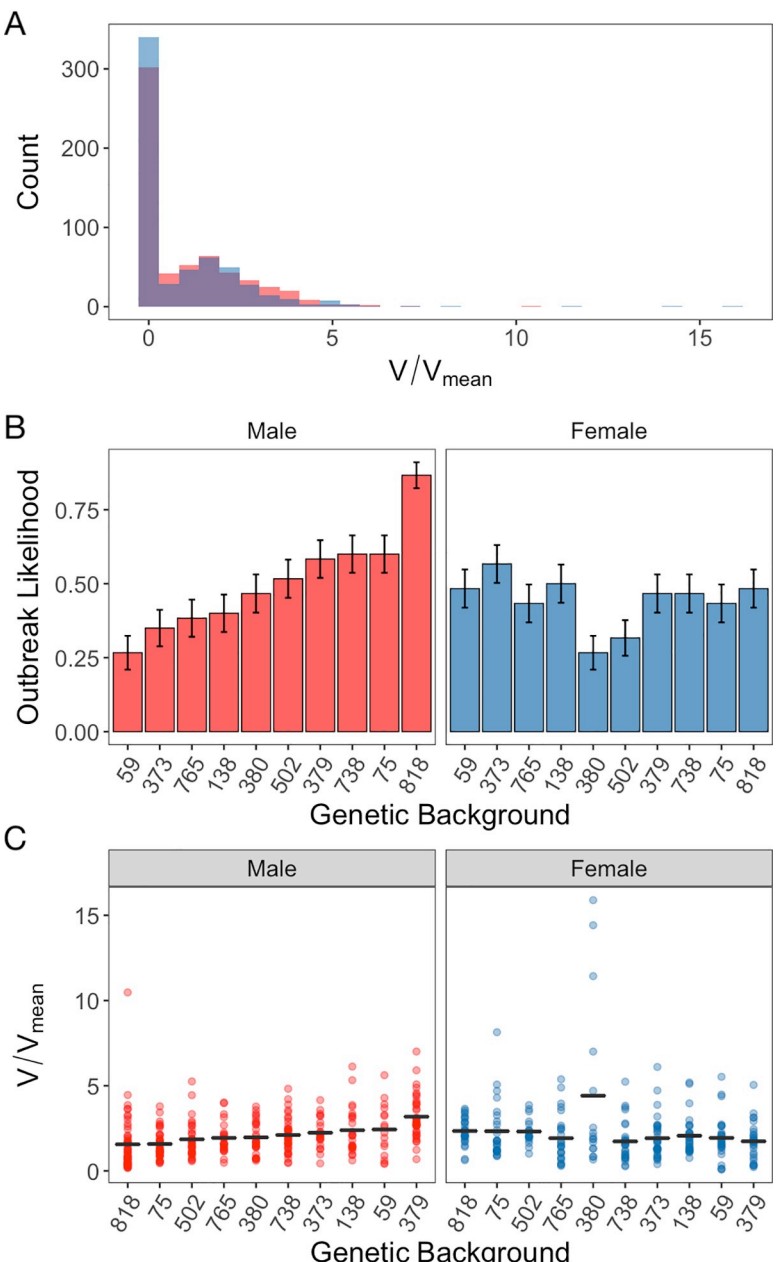

**Fig 4.** Bootstrap simulation results of transmission potential (*V*) (n = 60): (a) the distribution of *V* relative to the mean of the population ($V_{mean}$) in female (blue) and male (red) flies, overlap between these distributions is shown in purple. All individual transmission potentials are relative to the population average ($V/V_{mean}$ = 1). The mean±SE of (b) the proportion of flies with a non-zero transmission potential and (c) the transmission potential of flies with a non-zero transmission potential. In Figure. panels (b) and (c) sex is denoted by colour with males in red and females in blue. The x-axis of panels (b) and (c) is in ascending order of the male genetic backgrounds.

between the two (2.83%) explained relatively little deviance in the likelihood of an outbreak (Fig 5 and Table 7).

We then focused on the subset of flies with positive transmission potential *V*, to test which factors would affect the "quantitative" V, that is, in individuals where epidemics do take off (V>0), how big are these outbreaks and which factors explain variation in the size of the

**Table 7. Model outputs for the logistic regression analysis conducted on qualitative $V$ (the proportion of non-zero $V$ values).**

| Response Variable | Predictor | Df | $\chi^2$ | % Deviance Explained | p-value |
|---|---|---|---|---|---|
| Qualitative $V$ (Outbreak likelihood) | Sex | 1 | 4.58 | 0.28 | 0.032 |
| | Line | 9 | 38.2 | 2.30 | <0.0001 |
| | Sex*Line | 9 | 47.0 | 2.83 | <0.0001 |

outbreak? We found significant effects of genetic background and its interaction with sex explained which together explained over 15% of the variance in V (Fig 5 and Table 8).

## Discussion

We quantified genetic and sex-specific variation in three key determinants of DCV transmission: lifespan following infection, virus shedding, and virus load. When combined with social aggregation data, this variation resulted in significant effects of host genetics and sex on individual transmission potential, *V*. While all three traits influence transmission potential, virus shedding is of fundamental importance, and as in many individuals virus shedding was not detectable, this trait is likely to have a stronger contribution to the individual transmission potential *V* than variation in lifespan following infection or social aggregation. Below we discuss the central role of virus shedding to pathogen transmission, linking it to genetic and sex-specific effects we detected in *V* to the widely observed heterogeneity in pathogen spread.

### The effect of host genetic background in generating heterogeneity in transmission

The genetic background of the flies affected both whether flies were likely to shed DCV on a given day (which we called qualitative shedding), and also how much they would shed (quantitative shedding). Differences between genetic backgrounds in qualitative shedding was a key determinant of variation in *V*: in the absence of virus shedding there is no risk of pathogen transmission. Among individuals that shed DCV, between-individual heterogeneity in *V* was

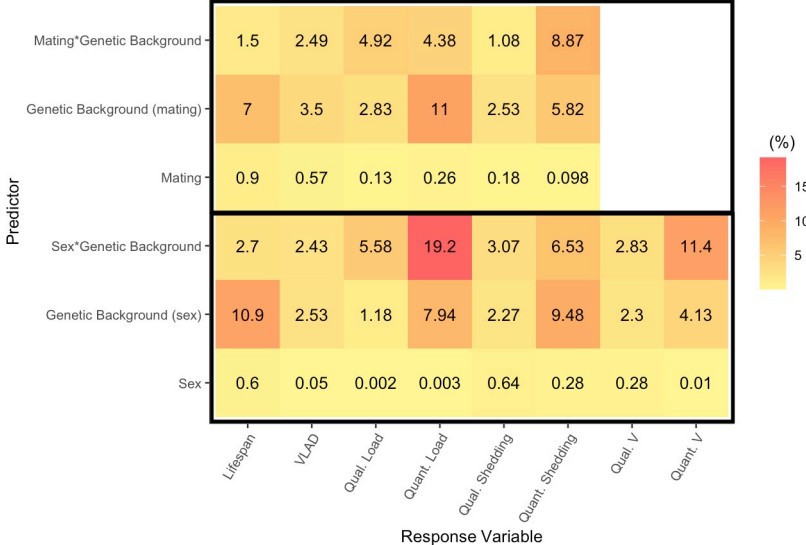

**Fig 5. Summary of the percentage of variance or deviance explained by a subset of predictors in analyses of disease transmission potential and outcomes of infection.** Separate analyses were used to test the effect of mating in females (top three rows) and sex (bottom three rows). We did not analyze the effect of mating on *V*, as this was not tested in the previously published social aggregation experiment [26] used to calculate *V*.

**Table 8. Model outputs for the GLM analysis conducted on quantitative *V* (the subset with non-zero *V* values).**

| Response Variable | Predictor | Df | F | % Variance Explained | p-value |
|---|---|---|---|---|---|
| Quantitative *V (Outbreak size)* | Sex | 1 | 0.077 | 0.01 | 0.78 |
| | Line | 9 | 2.51 | 4.13 | 0.008 |
| | Sex*Line | 9 | 6.94 | 11.4 | <0.0001 |

achieved through different routes. Some genetic backgrounds, such as males from RAL-818, showed a high proportion of individuals that are likely to spread DCV (Fig 4B), but show relatively low transmission potential during an outbreak (Fig 4C). Conversely, other groups, such as females of the RAL-380 genetic backgrounds, showed one of the lowest proportions of individuals able to achieve transmission (Fig 4B), but the individuals that did achieve transmission predominate the long-tail of the population's distribution with values of *V* that were several times higher than the population average (Fig 4C).

Host genetic background modified DCV shedding in distinct ways. Presenting detectable levels of viral shedding was affected by genetic background as part of an interaction with host sex, while this interaction has no significant effect on the quantitative variation in DCV shedding (Tables 5 and 6). Similar differences are seen in the amount of deviance and variance explained by genetic background in models of qualitative and quantitative variation in DCV shedding. Genetic background accounts for only 2.27% of deviance in qualitative DCV shedding whereas it accounts for 9.48% of the variance in quantitative DCV shedding (Fig 5). Genetic variation therefore appears to play an important role in determining shedding and affects qualitative and quantitative shedding in different ways. Similar effects of genetic backgrounds on parasite shedding have been reported in the Ramshorn snail species, *Biomphlamaria glabrata*, during infection with *Schistosoma mansoni*, where genetic backgrounds differ in how many parasite eggs they shed and how quickly they start shedding after infection [62].

Perhaps as a direct consequence of these host genetic effects on virus shedding, host genetic background also appears to play a key role in transmission potential *V*. The percentage of deviance and variance explained by genetic background does not hugely differ (2.3% and 4.13%, respectively). However, the interaction of genetic background with sex, accounted for 11.4% of the variance in quantitative variation in DCV shedding, whereas this same interaction only accounted for 2.83% of the deviance in qualitative variation in shedding (Fig 5). Put simply, genotype-by-sex interactions play a relatively small role in whether viral shedding occurs, but explain a larger proportion of the variance in how much virus is shed.

Genetic variation in pathogen shedding has been observed in a range of species and has been shown to have an important effect on transmission dynamics [43–45]. Families of turbot fish (*Scophthalmus maximus*) have been shown to produce outbreaks that differ in how quickly individuals show symptoms of infection and mortality rate. Despite not being measured directly, shedding is implicated at the heart of this variation, as no significant differences in *S. maximus* infection duration and contact rate have been found [63]. Spore shedding, specifically the number of spores released in the days after infection has also been shown to be affected by host genotype in the plant, *Plantago lanceolata*, when infected with multiple strains of the fungal pathogen, *Podosphaera plantaginis*, and this experimental data correlates well with differences in population-level transmission dynamics of wild *P. lanceolata* populations [43, 64].

## The effect of host sex in generating heterogeneity in transmission

Sex differences in transmission or virus shedding, lifespan and social aggregation are commonly observed in a wide range of species [65–68]. Our data adds to these body of evidence of

sex differences with clear qualitative and quantitative differences in $V$ between males and females. Other work has also shown a number of sex differences in pathogen and parasite shedding [68–70]. While the extent of any difference between males and females was also determined by the host genetic background, overall, males tended to have higher transmission potential $V$ when compared to females. Some of this sex effect is likely driven by higher virus shedding in male flies: males from several genetic backgrounds (RAL-379, RAL-738 and RAL-818) shed more DCV than females (Fig 3A). Interestingly, we observed significant sex-specific differences in the likelihood of detecting virus shedding, but not in the amount of virus shed. This suggests that males and females may differ in their ability to clear DCV infection altogether, but not in their ability to control titers if they are not able to clear infection. It is important to note however that, while the effect was significant, fly sex accounted for a very small percentage of the deviance in shedding (Fig 5).

Many examples of sex differences in behavioral and physiological traits that affect disease transmission have been observed across a range of host-pathogen systems [65]. Much of this work has been carried out in mammalian hosts, where the production of testosterone has been shown to promote transmission through increasing male contact rates [71–74]. Testosterone's influence on transmission heterogeneity likely extends further as its immunosuppressive effects [75] may also influence variation in infectiousness and infection duration. In non-mammalian systems, patterns of sex-specific variation in transmission are less clear [69]. Female-biased transmission is seen in the water flea, *Daphnia magna*, as females release significantly more parasite spores upon death than males [70]. Female *D. melanogaster* have also been shown to delay copulating with infected individuals longer than males, a behavior linked to their increased susceptibility[76].

## Female mating status in shedding

Mated and virgin females did not show any substantial differences, in almost all our measures of fly lifespan, viral load, viral shedding and transmission potential $V$. Two exceptions were a significant genotype-mating effect in the likelihood of having detectable titers of DCV, and in the amount of DCV these flies shed, suggesting that mating effects may only be present in some genetic backgrounds. This interaction between mating and genetic background explained ~5% of the variance in detectable DCV load and ~9% of the variance in the amount of DCV shedding (Fig 5). Alongside host genetic background, mating might exert a moderate influence over transmission there are differences between genetic backgrounds in post-mating physiological changes in the intestine that can increase defecation rates [77], which would likely lead to increased viral shedding. However, this physiological change would not explain why virgin females from particular genetic backgrounds shed more than mated females (Fig 3B). Relatively few studies have considered how mating affects aspects of disease transmission outside of contact rates [78, 79].

## The route of infection and detecting low-level virus shedding

An important caveat to the interpretation of our results is related to the septic route if infection we employed. DCV infection Is rare in wild flies [53] and its natural route of transmission is not entirely clear. It is very likely to be fecal-orally transmitted because it shows tissue tropism for the crop and midgut epithelia and is often found to be a lab contaminant in many stock lines[50, 54]. However, experimental oral infection has the limitation of requiring extremely high concentrations to establish a lethal infection[56, 80]. It would also introduce uncontrollable variation arising from differences between fly lines and sexes in their feeding rate. However, even during septic infection, DCV shows tissue tropism for the crop and gut, and given

that we detected DCV in fecal shedding, this indicates that it is viable to measure viral shedding following a systemic infection. However, the route of infection may explain to some extent why we found such a poor correlation between the internal DCV load and DCV shedding (S1 Fig). First, we were necessarily comparing the viral load in the whole fly to only the fraction of virus which established in the gut and was then excreted into the vial. A related consequence is that there is likely to be a time-lag between growth in the hemocoel, migration to the gut and finally externally shed virus. This time-lag between growth and shedding could also explain why we did not observe shedding in some individuals with elevated internal viral titers (super-sponges) while others presented undetectable internal loads but very high quantities of shed virus (super-shedders) (S1 Fig).

## Concluding remarks

By combining measures of virus shedding, lifespan and social aggregation into a simple framework for individual transmission potential, our work demonstrates that genetic and sex-specific variation can affect individual heterogeneity in different components of disease transmission. Our results are also consistent with the observation that the majority of infected individuals produce very few, if any, secondary cases of infection. Non-infectious individuals are particularly relevant to predicting outbreaks of infectious disease as they obscure high-risk individuals in traditional, population-wide estimations of outbreak risk. Finally, this work will hopefully also highlight the strengths of using model systems to systematically dissect the various sources of variation in pathogen transmission, which may serve as a tool for both hypothesis testing and hypothesis generating in disease ecology.

## Supporting information

**S1 Fig. The relationship between the viral load of flies and the amount of virus they shed into their environment.** The two distinct phenotypes, where individuals show a zero-value for shedding or load and a positive-value for the other trait, are marked by red (supersponges) or blue (supershedders).
(TIF)

**S1 Table. The number of flies measured for lifespan and viral load at death for each combination of genetic background and sex/female mating status.**
(DOCX)

**S2 Table The number of viral load samples for each treatment group (a) 1 DPI, (b) 2 DPI and (c) 3 DPI.**
(DOCX)

**S3 Table. Summaries of the logistic regression and GLMs used to analyse the response variables of our experiments.** All interactions are fully-factorial and marked using an asterisk (*).
(DOCX)

## Acknowledgments

We thank V. Gupta, K. Monteith, H. Borthwick, H. Cowan, and A. Reid for technical assistance and media preparation and F. Waldron for assistance with RNA extraction and qPCR troubleshooting. We are also grateful to M. Craft and L. White for in-depth discussion and enthusiastic support of the project.

## Author Contributions

**Conceptualization:** Pedro F. Vale.

**Data curation:** Jonathon A. Siva-Jothy.

**Formal analysis:** Jonathon A. Siva-Jothy.

**Funding acquisition:** Pedro F. Vale.

**Investigation:** Jonathon A. Siva-Jothy, Pedro F. Vale.

**Methodology:** Jonathon A. Siva-Jothy.

**Project administration:** Pedro F. Vale.

**Resources:** Pedro F. Vale.

**Supervision:** Pedro F. Vale.

**Visualization:** Jonathon A. Siva-Jothy.

**Writing – original draft:** Jonathon A. Siva-Jothy.

**Writing – review & editing:** Jonathon A. Siva-Jothy, Pedro F. Vale.

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
