## [Decision Letter · Decision Letter 0]

10 Jan 2020

Dear Dr Vale,

Your manuscript has now been reviewed by three well-qualified reviewers. Two reviewers with a primary EEID background appreciate the novelty of the work including the comprehensive nature of the measurements and the utility of the system. They suggest substantial revisions to the manuscript. The third reviewer has an EEID and pathogenesis background. They recommend rejection.

Overall, I think the reviews are fair in their appraisal and accurate in the details of their feedback. Because there is a wide range of opinion on the manuscript, it suggests that rewriting could improve the appeal to the PLOS Pathogens readership.

I think the comprehensive measurements and calculation of the determinants of inter-individual variation in disease transmission potential are valuable to the field and will enable new work by theoreticians and modelers. Furthermore, establishing such work in a genetics system such as Drosophila could lead to powerful future experiments. However, the reviewers also brought up methodological questions in terms of how the experiments were designed and executed. A discussion of the experimental logic and how it might affect the results would help. For instance, Reviewer 1 notes that the DCV virus is naturally acquired by feeding but here it is septically introduced. Could that be the cause of the high load, low shedding phenotype?

All three reviewers suggest improvements to the writing that could overcome some deficits. For instance, Reviewer 1 notes that the "lean" writing style could be improved by placing the methods section before the results. Reviewer 2 makes many helpful suggestions as to how the authors could clarify the writing, which would also address some of the concerns of Reviewer 3. Reviewer 3's primary concern was a lack of novelty. More specific clarification of the gaps in our knowledge in the abstract and introduction should be made in order to convey the importance of this work to those outside the field of EEID.

I encourage you to substantially address all of the reviewer concerns and resubmit. In your rebuttal letter, please be sure to carefully address all of the reviewer feedback.

Generic text follows....

Thank you very much for submitting your manuscript "Dissecting genetic and sex-specific sources of host heterogeneity in pathogen transmission potential" (PPATHOGENS-D-19-02043) for review by PLOS Pathogens. Your manuscript was fully evaluated at the editorial level and by independent peer reviewers. The reviewers appreciated the attention to an important problem, but raised some substantial concerns about the manuscript as it currently stands. These issues must be addressed before we would be willing to consider a revised version of your study. We cannot, of course, promise publication at that time.

We therefore ask you to modify the manuscript according to the review recommendations before we can consider your manuscript for acceptance. Your revisions should address the specific points made by each reviewer.

(1) A letter containing a detailed list of your responses to the review comments and a description of the changes you have made in the manuscript. Please note while forming your response, if your article is accepted, you may have the opportunity to make the peer review history publicly available. The record will include editor decision letters (with reviews) and your responses to reviewer comments. If eligible, we will contact you to opt in or out.

(2) Two versions of the manuscript: one with either highlights or tracked changes denoting where the text has been changed; the other a clean version (uploaded as the manuscript file).

Additionally, to enhance the reproducibility of your results, PLOS recommends that you deposit your laboratory protocols in protocols.io, where a protocol can be assigned its own identifier (DOI) such that it can be cited independently in the future. For instructions see http://journals.plos.org/plospathogens/s/submission-guidelines#loc-materials-and-methods

We hope to receive your revised manuscript within 60 days. If you anticipate any delay in its return, we ask that you let us know the expected resubmission date by replying to this email. Revised manuscripts received beyond 60 days may require evaluation and peer review similar to that applied to newly submitted manuscripts.

[LINK]

Sincerely,

William B. Ludington, PhD

Guest Editor

PLOS Pathogens

Sara Cherry

Section Editor

PLOS Pathogens

Kasturi Haldar

Editor-in-Chief

PLOS Pathogens

orcid.org/0000-0001-5065-158X

Michael Malim

Editor-in-Chief

PLOS Pathogens

orcid.org/0000-0002-7699-2064

Reviewer's Responses to Questions

**Part I - Summary**

Reviewer #1: This study examines the contributions of host sex, mating status, and genetic background to among-host variation in the transmission potential of Drosophila C Virus. The study finds that some of these variables and their interactions account for appreciable variance in transmission-relevant traits like quantitative resistance, viral shedding, and lifespan post infection. The variance in these traits is further used to bootstrap and estimate the distribution of transmission potential among individuals in a population, finding a combination of zero-inflation and a small but potentially epidemiologically important superspreader tail.

The motivation for this study is excellent – the field desperately needs work that rigorously connects variation in the within-host dynamics (e.g. viral load, qualitative vs quantitative resistance) of infection to between-host dynamics, and this study ambitiously approaches this goal. Thus, I think it is of sufficient novelty and broad interest for the readers of this journal.

I struggled in my first reading of this manuscript because it is written in an excessively lean style, given that the methods do not appear till the end. Several key “what” and “why” details should be elaborated upon in the results and discussion, to help readers better understand the experimental design and relevance and limitations of the results (specific suggestions below).

Reviewer #2: This manuscript aims to explore how three factors of host identity (sex, genetic background, and mating status) interact to produce heterogeneity in pathogen transmission potential. Using a model host-pathogen system (Drosophila and DCV), they found extensive among-individual variation in transmission potential, and a much of this variation is explained by genotype x sex interactions. The experiments are exploratory, but provide interesting empirical data on transmission heterogeneity, and the authors combine these measures with previous data on fly social behavior to estimate V (individual transmission potential) and present these data very well (e.g., Figure 4a). Overall, the intellectual merit of this paper is twofold: the authors describe and quantify potential factors contributing to pathogen transmission heterogeneity and their interactions, and they also provide a great foundation for future studies using this model host-pathogen system. Drosophila is clearly recognized as a model for physiology, genetics, and pathology, but these types of studies show how powerful this system can be used as a tool to study ecology and evolution of infectious diseases. I think this will make a fantastic contribution to the literature, and will garner broad readership in PLoS Pathogens.

Reviewer #3: The manuscript by Jc fonathon A. Siva-Jothy and Pedro F. Vale has tested how common sources of variation between individuals (genetic background, sex and mating status) contribute to individual heterogeneity in pathogen transmission potential using Drosophila melanogaster and Drosophila C Virus as a model system.

The primary weakness of the paper, conceptually and in terms of novelty, is that the effect of host trait on disease transmission has been determined in other host-pathogen system. Indeed, the authors here systemically measure the effect of multiple host traits on disease transmission, which provides new information to fully understand the sources of heterogeneity in pathogen transmission. Overall the paper is succinct and a thorough study of the effect of host trait variation on pathogen transmission.

However, I have some lingering questions after reading the paper and suggestions for experiments that could improve or substantiate the work presented here.

**Part II – Major Issues: Key Experiments Required for Acceptance**

Reviewer #1: I also have three intermediate-to-major concerns about the experimental design that at least deserve substantial discussion or clarification, if additional supporting data is not readily achievable:

1) I understand that septic infection is convenient in that you can closely control the initial dose (a key source of variation), but this is a fecal-oral virus and you are measuring fecal transmission, correct? Given what is known about the life history of this virus, might you expect different results (e.g. the degree to which different variables and their interactions explain transmission virulence) if infected through the natural transmission mode? Most importantly, would you still predict the same zero-inflated + superspreader distribution? I think the absence of data on this dampens the impact of this paper (but maybe I am missing something about the host-microbe model here).

2) The intra-fly correlation between lifespan and viral shedding is not clear, because very few flies (even those with measurable viral loads) die before 10 days, but shedding is only measured in the first three days. Viral Load at Death is measured when the flies die, but there is no significant correlation between viral load and viral shedding (supp. figs). So, how does variation in lifespan really speak to the transmission period in this model? If this was carefully considered, it is not clear from the text.

3) Is the estimated heterogeneity in transmission potential really reflective of a natural population? Fig. 4 distribution shows a lot of variance, but it is worth noting that the data are from the 5 most resistant and susceptible Dmel lines, meaning that the variance has been artificially maximized. The distribution in a natural heterogeneous population is likely much lower.

Reviewer #2: 1. I think the term “superspreader” should be de-emphasized in the introduction (and removed from the keywords). Although superspreader events are an extreme form of transmission heterogeneity, this paper does not provide any direct empirical evidence for how superspreader events can be generated by differences in host traits. I think that future studies could certainly use this system to do so, and this paper certainly sets the groundwork for those experiments.

2. The abstract might be more meaningful if the authors included some quantitative descriptors of their results, perhaps some of the variance explained by sex, genotype, mating status, or the effect size/magnitude of change found between sexes or the most different genotypes. Saying that one genetic background had X% higher viral shedding, for example, than another is more informative than just saying there are vast differences between genotypes.

3. I suggest that the authors be more forthright in the introduction that this is exploratory research aimed at discovering the degree of heterogeneity in pathogen transmission potential, rather than testing hypotheses about specific genotypes, sexes, or their combinations. Given the huge amount of work involved in these experiments, this isn’t a criticism by any means, this type of exploratory work is very important, but the results will be put in a clearer context if the reader knows these aims by the end of the introduction.

4. It would be helpful if the authors expounded a bit on the difference between qualitative and quantitative values of shedding. I understand how you measured these things, but what they mean in the context of biology is still unclear.

5. Since one of the goals of the paper is to show how this system is a valuable model for studying transmission heterogeneity, I would have liked to see some more information about the natural history of the interaction between Drosophila and DCV, if there is anything known about prevalence in the wild, evidence for epizootics, etc.

Reviewer #3: 1. It has been reported in many host-virus systems that genetic and sex-specific variation can affect individual heterogeneity in different components of disease transmission, which is interpreted again in this study, my major concern of this study is the lack of novelty.

2. In the main text, only mated females were measured, there is no explanation for not characterizing mated males.

3. Figure 2a: do you have any ideas that the proportion of flies with non-zero DCV load differed between virgin and mated females from particular genetic background? It should be somehow discussed.

**Part III – Minor Issues: Editorial and Data Presentation Modifications**

Reviewer #1: Other Comments

Lines 105-107: More rigorous examples where covariance among traits (other than just the anecdotal typhoid Mary and SARS) would help to underscore the existence of a gap in knowledge here. Also, what is “coupled heterogeneities?” That sounds like a term that I would use as filler if I didn’t know quite what I wanted to say. I think it needs clarification.

Lines 107-108: I would change "fully" and "essential" in this sentence; measuring multiple traits may still not allow you to fully understand, and conversely a full understanding may not require the measurement of multiple traits (perhaps one trait explains an outsized amount of variance)

Lines 117-120: I think it is essential here that you note which measurements come from the same individual flies (e.g. allow you to get to the intra-fly covariance) and which are relying on averages among fly groups. For better or for worse, the impact (and the most important limitations and reservations, to my mind) come from the extent to which these measurements are coupled across time and within flies.

Intro: I think it would be helpful somewhere in here to introduce the concept of quantitative vs. qualitative infection outcomes (including biological examples of systems where this is an important consideration), since it becomes so central to the analysis and interpretation of the results.

Results: Overall, the results needs more of an introduction to the general experimental design. How is oral infection carried out? How many genotypes did you infect? How long does it take flies to die, in general? When did you measure viral load? If the methods come at the end of the document, these details need to be here, to provide necessary narration. Otherwise just put the methods before the results; surely PLoS has some leeway there.

Lines 126-127: Some summary statistics would be helpful here. On average, how much did infection reduce lifespan?

Fig 1A: Kind of hard to eyeball general trends across columns here, other than they aren't precisely rank ordered

Line 133: Interesting. Is this because later-dying flies started clearing infection and then died, or do you think that cumulative load, rather than peak load, explains mortality?

Line 136: Were they not properly infected/colonized or did they clear infection? This section would benefit from a sentence or two introducing the experiment in question, since the methods are at the end of the document

Line 140: Should this be labeled as “detectable” instead of ‘non-zero’ on the Fig. 2a axis?

Line 141: These are collected in first three days after infection. Do flies start to die before this? Do they start to die well after this? What does DCV titres at this stage really tell you?

Line 141; Fig. 2a: Not clear from just looking at the figure that those bars are time points. Why not represent the data in the same way that you represent the quantitative data, or show more of a binomial style of plot with actual data points?

Line 149-150: Why are these pairwise? Is this better than a quadratic model?

Line 156: How is shedding measured? Give us a crumb of a method here.

Line 176: did you use individual distribution over time curves for this?

Line 180: Fig. 4a distribution is interesting. What does R0 mean for a proportion? Not quite sure I get the x axis there.

Line 180, and Line 293: These paragraphs need some substantial revising here and there for grammar

Line 208-9: Do those that fail to shed also survive? If not, could there be shedding post mortem?

Lines 235-61: This paragraph is long and starts to get a little list-like

Line 267: Would it be worth it to show male vs female V/Vmean distributions, perhaps as overlapping in Fig. 4A?

Lines 281-288: I appreciate the connection to other systems, but the testosterone discussion roams pretty far from being relevant for your results.

Lines 289-90: This example just hangs there. Are you trying to make a point about vertebrate vs. invertebrate sex bias?

Lines 312-313: Do we know what that dose response looks like? Is it threshold-like? Could R0 actually be lower than estimated?

Lines 318, and surrounding: I think there needs to be a deeper and more thoughtful discussion of how sampling limitations and within-group variation reflect on covariance among variables

Lines 340-1: Which 5 and 5 genotypes? Did your results recapitulate this?

Line 362 and Line 373: This is not a huge sample size, given the clear variation in clearance rates and zero-inflation. Probably sums up to a good sample size for single variables, but the interaction effects might be underpowered (e.g. some genotype x trait groups have an N=1). Also, was the experiment repeated multiple times? Where do the error bars come from?

Line 370: Did you have a group of flies that were measured for only load without the shedding assay first, to see if it was stressful and/or made a difference for the load results?

Fig S1: I think Fig. S1 labels are backwards - shouldn't high load but low VLAD individuals be supersponges?

Reviewer #2: 6. Lines 14-15: The abstract sets up the paper as if you will measure the se underlying effects, but rather you manipulate host genotype/sex/mating status and find widespread effects on transmission potential. However, the mechanisms remain unknown. An extra sentence in the abstract to explain that factors of host identity like genetic background and sex can influence these underlying mechanisms that then influence transmission potential.

7. Line 88: Since immune traits are a form of physiological traits, I would replace that with “social factors” or something to point out that transmission is influenced by the interaction between infected hosts and susceptible hosts. A relatively recent paper showed that social context can be a predictive factor of disease risk in Drosophila social aggregations:

Keiser, C.N., Rudolf, V.H., Sartain, E., Every, E.R. and Saltz, J.B. 2018. Social context alters host behavior and infection risk. Behavioral ecology, 29(4), pp.869-875.

8. Line 141: DPI should be defined before its first usage here, as well as what you mean by “DPI effect”.

9. Line 182: Ah, here we see some evidence for potential superspreaders. “Long-tail” distributions like this are common in many diseases, and the individuals are the far right end may represent superspreaders (as described in Lloyd-Smith’s Nature paper from 2005). If the authors prefer to keep their references to superspreaders in the introduction (see my comment #1), then this should be highlighted in the discussion more explicitly.

10. Line 185: Statements like this, and throughout the results section, would be much more powerful when they include some effect size – what was the sex-difference in proportion here?

11. Lines 242-261: This seems like a laundry list of studies that showed genotypic effects on pathogen shedding/transmission – it might read a little easier if they were distilled into main take-away messages, and how they help explain the effects discovered in your experiments.

12. Line 267: How is this statement different from the opening sentence of this paragraph?

13. Line 279: There are also studies on Drosophila sex-differences in pathogen avoidance, and how that is related to susceptibility (e.g., Keiser et al 2019, Sex differences in disease avoidance behavior vary across modes of pathogen exposure; Ethology; and references therein). How avoidance behavior related to transmission potential might be an interesting addition to this paragraph.

Reviewer #3: 1. The paper has used extensive long sentences without the proper commas resulting in difficulties to follow. It can be simplified by reducing the use of complex sentence structures.

2. Figure 1a&b: statistics should be noted for the two figures either in main text or in the legend.

3. Figure 2a&b: statistics should be note, and mark for significance should be denoted.

4. Carefuly check the writing, such as in Line 161, please carefully check ‘while these were effects were significant’.

5. The authors should carefully check the format of reference and keep it consistent, for example, for some journal name, abbreviations were used. The paper also excessively cites publications; less than half of the citations are likely necessary.

6. Line 314, the format of the two dashes is different.

PLOS authors have the option to publish the peer review history of their article (what does this mean?). If published, this will include your full peer review and any attached files.

Reviewer #1: No

Reviewer #2: No

Reviewer #3: No

---

## [Decision Letter · Decision Letter 1]

30 Nov 2020

Dear Dr Vale,

We are pleased to inform you that your manuscript 'Dissecting genetic and sex-specific sources of host heterogeneity in pathogen shedding and spread' has been provisionally accepted for publication in PLOS Pathogens.

Best regards,

William B. Ludington, PhD

Guest Editor

PLOS Pathogens

Sara Cherry

Section Editor

PLOS Pathogens

Kasturi Haldar

Editor-in-Chief

PLOS Pathogens

orcid.org/0000-0001-5065-158X

Michael Malim

Editor-in-Chief

PLOS Pathogens

orcid.org/0000-0002-7699-2064

Congratulations on acceptance of your manuscript. We think the paper will be a classic and will motivate others to also assess the role of heterogeneity in individual transmission potential in viruses using model organisms. The work is pioneering and the topic is of course highly relevant to global health.

For the revised draft, the first two reviewers were asked whether the revisions met their requirements. These two and the guest editor had consensus that the revisions appropriately addressed all concerns.

Again, congratulations. This is very exciting work.

Two notes for final revision:

There is one minor typo:

ln 27: measures  measures of

Also, on line 167: please very briefly explain why flies were reared at low density, e.g. "to provide optimal food availability." or "to minimize any chance of any unknown, endogenous viral infection prior to start of the experiment."

Reviewer Comments (if any, and for reference):

Reviewer's Responses to Questions

**Part I - Summary**

Reviewer #1: In the revised version of the manuscript, the authors have thoughtfully addressed my previous concerns. In particular, they provide a reasonable justification for performing septic rather than oral infections, and (thanks to improved organization and explanation of methods) it is much easier to follow the analyses and comparisons of individual metrics and calculations/distributions of transmission potential. The resulting manuscript is a pleasure to read, the model system is likely to inspire future work, and the results should be of substantial interest to those interested in disease ecology, heterogeneity, and host-microbe interactions. I have no further concerns about the suitability of this manuscript for publication.

Reviewer #2: The authors have sufficiently addressed all of my comments. Great job!

**Part II – Major Issues: Key Experiments Required for Acceptance**

Reviewer #1: None

Reviewer #2: (No Response)

**Part III – Minor Issues: Editorial and Data Presentation Modifications**

Reviewer #1: One tiny thing: there appears to be a typo in Line 27 of the abstract – probably important to fix.

Reviewer #2: (No Response)

PLOS authors have the option to publish the peer review history of their article (what does this mean?). If published, this will include your full peer review and any attached files.

Reviewer #1: No

Reviewer #2: No

---

## [Editor Report · Acceptance letter]

13 Jan 2021

Dear Dr Vale,

We are delighted to inform you that your manuscript, "Dissecting genetic and sex-specific sources of host heterogeneity in pathogen shedding and spread," has been formally accepted for publication in PLOS Pathogens.

Best regards,

Kasturi Haldar

Editor-in-Chief

PLOS Pathogens

orcid.org/0000-0001-5065-158X

Michael Malim

Editor-in-Chief

PLOS Pathogens

orcid.org/0000-0002-7699-2064